# Physiological and Putative Organic Cation Transporter Expression Response to Alizarin Dye Exposure in *Aedes aegypti* Mosquitoes

**DOI:** 10.3390/insects16121196

**Published:** 2025-11-25

**Authors:** Naomi R. Kennel, Matthew F. Rouhier

**Affiliations:** Department of Chemistry, Kenyon College, Gambier, OH 43022, USA; kennel.naomi@gmail.com

**Keywords:** xenobiotic transport, *Aedes aegypti*, vector-borne illness, qPCR

## Abstract

Mosquitoes threaten 40% of the world’s population with the pathogens they spread. Though many attempts to control mosquitoes have been tried in the past, alternative and more sustainable control strategies are still needed. Here, we investigated the way mosquitoes remove foreign molecules by injecting mosquitoes with synthetic dyes in saline. Next, we collected the urine and measured how much of the dye was removed by the mosquito. Lastly, we looked at how a small group of transport proteins, believed to help in the removal of foreign molecules, changed in expression following dye exposure. This work provides insights that could inform new molecular targets for mosquito control.

## 1. Introduction

Mosquito-borne illness is an increasing challenge to global health, threatening over 40% of the world’s population [1]. More specifically, *Aedes aegypti* mosquitoes are the primary vectors of many pathogenic viruses that can be transmitted to humans, such as dengue, chikungunya, Zika, West Nile virus, and, most notably, yellow fever. Presently *Ae. aegypti* are found in Africa, South and Central America, southern North America, the Middle East, Southeast Asia, the Pacific and Indian Islands, northern Australia, and sporadically in Europe [2]. Due to their synanthropic adaptation, they are more likely vectors for these pathogens than other types of mosquitoes [3]. Climate change will exacerbate the risk of vector-borne illness as a large contributor to the global burden of disease [4].

Previous efforts to control the spread of vector-borne illness have relied heavily on the regular use of chemical insecticides, which leads to resistance among vectors, environmental hazards, and adverse effects on public and animal welfare [2]. On the other end of the spectrum, environmental management techniques are safe for humans and non-target organisms but are costly, variably effective, and difficult to manage due to their operational complexity, high cost, and high maintenance for implementation and vector surveillance [5]. As a result, there remains an urgent need to develop novel control methods that are economic, biodegradable, environment-friendly, and safe to non-target organisms in order to combat the mosquito-borne illness [6].

An alternative strategy is to study molecular mechanisms underlying insect detoxification and target them instead. More specifically, the inhibition of xenobiotic transporters can increase mosquito vulnerability to insecticides. For example, a study investigated ATP-binding cassette (ABC) transporter expression following permethrin exposure and found that ABC transporter inhibition increased *Anopheles gambiae* larval mortality by about 15-fold [7]. Another study used dsRNA-mediated gene silencing to successfully decrease chitin synthase A (CHSA) mRNA levels, consequently decreasing *Ae. aegypti* survival [8].

With the development and eventual implementation of these tactics, there is an ever-growing need for knowledge regarding the molecular and genetic mechanisms in *Ae. aegypti*. However, many xenobiotic transporters, such as novel organic cation transporters (OCTNs), are uncharacterized in *Ae. aegypti,* even though their transport capabilities are likely very important in many mosquito processes. There is a need to identify these transporters and elucidate their natural substrates and mechanisms. Therefore, the goal of this work is to use quantitative PCR to measure the expression of predicted *Ae. aegypti* OCTNs at two time points after volume-challenging with Alizarin Yellow GG, Alizarin Yellow R, or Olsalazine to stimulate diuresis. We characterized the expression at 2 and 24 h post volume-challenge to capture a snapshot of what transport processes occur in the initial diuretic response and the subsequent recovery/processing phase. These time points are physiologically relevant as a mosquito’s response to a blood meal fluctuates 0–72 h post blood meal [9]. Based on the patterns of gene expression, the possible function of these proteins can be explored. We hypothesized that OCT(N)s are involved in xenobiotic clearance and that exposure to alizarin dyes would alter their expression.

## 2. Materials and Methods

### 2.1. Identification of Putative Organic Cation Transporters in Aedes aegypti

To search for putative organic cation transporters, we began by assembling a complete set of human SLC22A protein sequences. We then utilized the BLAST (v 2.14.0) function of VectorBase [10] to query the *Ae. aegypti* proteome with each individual human SLC22A sequence. The returns frequently clustered into two groups: those with Expect values (e-values) less than 1 × 10^−30^ and those with values nearer to 1. Individual returns with an e-value ≤ 1 ×10^−30^ were then further evaluated for identities of >25% and positives of >45% (the identity of distant human SLC22A proteins), a sequence length of 525 amino acids ± 50 (typical of common human isoforms), and were predicted to contain transmembrane spanning regions before being selected for alignment. The candidate sequences were then aligned with several programs (Clustal Omega: v 1.2.4 [11], COBALT [12], DiAlign [13], Kalign [11], MAFFT [14], MSAProbs [15], MultAlin [16], Phylogeny_fr [17], PROMALS [18], PSI-TM-Coffee [19], and PRANK [20]), but the alignments were inconsistent between programs. Therefore, we utilized CCTOP [21] to identify the transmembrane spanning regions and truncated the sequences to only include those from transmembrane region 1 to region 12. The alignment programs were run with the truncated sequences, and a consensus was reached. PEPT5 was included as a control. An unrooted phenogram of the alignment was prepared using Interactive Tree of Life (iTOL) [22].

### 2.2. Aedes aegypti Colony

*Aedes aegypti* were reared from eggs obtained through BEI Resources, NIAID, NIH: *Ae. aegypti*, Strain LVP-IB12, Eggs, MRA-735, contributed by David W. Severson. Batches of eggs were hatched by applying a vacuum, and each hatch contained approximately 150–200 eggs. Post-hatching, larvae were fed finely ground Tropical Tetrafin fish flakes (Blacksburg, VA, USA), which were supplemented with yeast extract (4:1 by mass) for the first two days and non-supplemented fish flakes for the remainder of their aquatic stage. Emergent pupae were transferred in a beaker to a collapsible mesh cage (BugDorm, MegaView Science, Taiwan). Adult mosquitoes were fed 10% (*w*/*v*) sucrose/water ad libitum. At all life stages, *Ae. aegypti* were maintained at 28 °C in an environmental chamber with a maintained humidity of 70–80% and a 14 h light/10 h dark cycle.

### 2.3. Mosquito Injections

Five-to-ten-day-old adult female *Ae. aegypti* were cold anesthetized on ice in small batches. Injections were performed using a capillary needle (Drummond Scientific Company—Broomall, PA, USA—part number 3-000-203-G/X) through the metathoracic spiracle with a Nanoject III (Drummond Scientific Company—Broomall, PA, USA). Final injection solutions consisted of 5% Dimethyl Sulfoxide (DMSO) (Fisher BioReagents, Waltham, MA, USA) in 1× Phosphate-Buffered Saline (PBS), diluted from a 10× solution (Fisher BioReagents, Waltham, MA, USA). The DMSO aided in the dissolution of the dye molecules. Dye stock solutions were prepared at concentrations of 50 mM and then diluted in PBS to 2.5 mM. The working concentration was chosen from preliminary injection trials showing minor toxicity at 2.5 mM. The dyes utilized for injections were Alizarin Yellow GG (Allied Chemical Corporation, New York, NY, USA), Alizarin Yellow R (Santa Cruz Biotechnology, Dallas, TX, USA), and Olsalazine Sodium (Apexbio, Houston, TX, USA). All solutions were filtered with a PVDF 0.22 μm sterile filter before injection (Restek Corporation—Bellefonte, PA, USA). Two injection volumes were utilized, 100 nL to establish toxicity and 900 nL to stimulate diuresis. All experiments were repeated with a minimum of 3 biological replicates.

### 2.4. Dye Toxicity Assay and Urinalysis

Ten female mosquitoes were injected with 100 nL as described above, transferred to a recovery container, and placed into the environmental chamber. The recovery containers were supplied with a vial of 10% sucrose. Mortality was assessed 18–24 h post-injection; mosquitoes were considered dead if they failed to move upon agitation.

For urinalysis, mosquitoes were injected with a 900 nL bolus to simulate diuresis. After injection, the mosquitoes were placed immediately in a graduated, packed-cell volume tube (MidSci, St. Louis, MO, USA)—5 mosquitoes per tube—and held at 28 °C. At two hours post-injection, the mosquitoes were transferred to a recovery container, and the accumulated urine was pooled into the graduated portion of the tube, and the volume was recorded. This tube was then inverted over a collection apparatus leading into a PCR tube and was spun down at 1000× *g* for 1 min at room temperature into the PCR tube using a Sorvall Legend X1R Centrifuge (Thermo Fisher Scientific, Waltham, MA, USA). All mosquitoes were confirmed to be alive at the end of the two hours, at which point ½ of the mosquitoes were sacrificed to analyze gene expression (whole-body tissue—no head, legs, or wings—suspended in 200 μL TRIzol^®^ reagent (Zymo Research, Irvine, CA, USA) and then stored at −80 °C). The remaining mosquitoes were returned to the environmental chamber and assessed for mortality (dye toxicity) at 24 h before being sacrificed for gene expression. All the statistical analyses were performed with GraphPad Prism 10.6.1 Software (GraphPad Software, San Diego, CA, USA), and this software was also used to generate graphs.

### 2.5. Measurement of Dye Clearance

The collected urine was also examined for dye clearance. Quantification of the dye was performed by collecting the absorbance spectra of the voided dye with a NanoDrop One Spectrophotometer (Thermo Scientific, Waltham, MA, USA) and comparing them with prepared standards. The molar absorptivity values were calculated using a concentration range of 0.0025 M to 0.8 mM in PBS. The mosquito urine was diluted two-fold in PBS to adjust for the pH dependence of absorption. Olsalazine was diluted four to eight-fold due to a greater initial concentration. The molar absorptivity values are 14.2 L mol^−1^ cm^−1^, 22.2 L mol^−1^ cm^−1^, and 29.6 L mol^−1^ cm^−1^ for Alizarin Yellow R, Alizarin Yellow GG, and Olsalazine, respectively, on the NanoDrop One Spectrophotometer (Appendix A).

### 2.6. RNA Extraction and cDNA Synthesis

All tissue samples were suspended in TRIzol prior to storage at −80 °C. They were thawed on ice and homogenized using zirconium oxide beads (Next Advance, Inc., Try, NY, USA) and a SPEX SamplePrep MiniG 1600 tissue homogenizer for 3 min at 1500 rpm and at room temperature. The RNA was extracted using a Monarch^®^ Total RNA Miniprep Kit (New England Biolabs, Ipswich, MA, USA), including a DNase I treatment step following the manufacturer’s guidelines. RNA was cleaned using an RNA Clean and Concentrator kit (Zymo Research, Irvine, CA, USA) with an additional DNase I treatment following the manufacturer’s guidelines. RNA was reverse transcribed to cDNA using the Verso cDNA Synthesis Kit (Thermo Fisher, Waltham, MA, USA) using random hexamers according to the manufacturer’s protocol.

### 2.7. Primer Design

The RealTime PCR Tool (Integrated DNA Technologies, Coralville, IA, USA) was utilized for primer design based on mRNA sequence data on VectorBase [10]. All default parameters were maintained; amplicon length of 70–150 bp, primer Tm 60–64 °C, and primer pairs spanning an exon-exon junction to enhance transcript specificity [9]. An RPS17 and ACT combination served as endogenous controls based on previous qPCR studies on housekeeping genes in *Ae. aegypti,* where ACT/RPS17 provided the best normalization of *Ae. aegypti* samples across many conditions [23]. RPS17 and ACT primers were designed based on this study [23], but RPS17 was found to produce multiple products in our preparations. Therefore, a second set of RPS17 primers was designed and ultimately implemented. A full list of primer and amplicon information can be found in Appendix A.

To ensure primer function and specificity, a normal workflow was performed until post-cDNA synthesis, where semi-quantitative PCR was performed using GoTaq^®^ Green Master Mix (Promega Corporation, Madison, WI, USA). A reaction of 50 μL was halted at 15, 20, 25, and 30 cycles to withdraw a 10 μL aliquot. Products were visualized on a 1.5% agarose gel to determine if product amplification was specific (single amplicon of the correct size) and efficient (bands would grow in intensity from 15 to 30 cycles) for each target transcript. From this preliminary testing, the primers listed in Supplemental Appendix A were selected for quantitative PCR.

### 2.8. Real-Time qPCR

Real-time quantitative PCR was performed using 2× PowerUp™ SYBR™ Green Master Mix (Thermo Fisher Scientific, Waltham, MA, USA) and following the instructions in the user guide for a 10 uL reaction. Reactions were run on an Applied Biosystems QuantStudio 3 Real-Time PCR System. Prepared samples included PowerUp SYBR Green Master Mix with 1.5 ng of cDNA per reaction (cDNA reaction contained 2.5 ng/μL of RNA) and 500 nM each of sense and antisense primers. Each qPCR reaction was paired with a no reverse transcriptase control (No RT) and a no template control (NTC) to detect genomic or outside contamination. All reactions were run in technical triplicate.

All primer combinations show 90–110% efficiency at this primer concentration and this cDNA concentration (RNA equivalents). A melt cycle was performed to analyze the products of the qPCR (see Appendix A for a full account of the thermocycle), as well as running the samples on an agarose gel to ensure the amplicons matched the predicted sizes (Appendix A).

Ct values were analyzed using Double Delta Ct Analysis, which assumes that there is equal primer efficiency between primer sets, that there is near 100% amplification efficacy of the reference and the target genes, and that the internal control genes are constantly expressed and are not affected by the treatment [24]. Statistical significance was determined by one-way ANOVA comparing the difference between the average fold change value for each time point to 2^−ΔΔCt^ = 1 for mosquitoes injected with PBS and harvested after 2 h.

## 3. Results

### 3.1. Identification of Putative Genes for Xenobiotic Transporters

Our broad screening for SLC22A-like xenobiotic transporters in *Ae. aegypti* revealed 14 candidate genes. Among the 14 genes they aligned into three groups, 1 with the Organic Cation Transporters (*AAEL004451*), 5 with the Novel Organic Cation Transporters (*AAEL000902*, *AAEL012443*, *AAEL024953*, *AAEL026837*, and *AAEL004479*), and 8 aligned to form a mosquito-specific group (*AAEL009206*, *AAEL011275*, *AAEL22769*, *AAEL013271*, *AAEL024914*, *AAEL003148*, *AAEL003192*, and *AAEL013489*) (see Figure 1). Notably, none of the genes aligned with the Organic Anion Transporters. All candidate transporters contained 12 predicted transmembrane regions, cysteine residues in the TM1-TM2 loop, and conserved motifs essential for function (Appendix A). We selected from among the transporters the groups of genes aligning with human OCT and OCTN for gene expression.

### 3.2. Establishing a Set of Xenobiotics for Gene Expression

The initial step in evaluating changes in xenobiotic transporter expression was to establish a set of xenobiotics that are neither too toxic nor too benign; we chose the dyes Alizarin Yellow GG, Alizarin Yellow R, and Olsalazine (Figure 2—top). All three dyes contain a common azo benzene core but have varied chemical moieties to potentially trigger different physiological responses. Alizarin Yellow GG and Alizarin Yellow R have a nitro group on their benzene rings at meta and para positions, respectively. Olsalazine has a carboxyl group in the meta position and a hydroxyl group in the para position.

To evaluate their toxicity toward mosquitoes, injections were performed with 100 nL of the 2.5 mM dyes and compared by one-way ANOVA (Figure 2—bottom). At 24 h, the average survival for PBS-injected mosquitoes was 92.0% with a standard deviation of 11.0. Alizarin GG was 88.0% with a standard deviation of 8.4, Alizarin R was 82.0% with a standard deviation of 11.0, and for Olsalazine, 84.0% with a standard deviation of 11.4. There were no significant differences in the mortality of 100 nL injection groups after 24 h; therefore, it was concluded that this volume and concentration were not overtly toxic to the mosquitoes.

Next, mosquito injections were performed with 900 nL of 2.5 mM dyes. After 2 h, all mosquitoes were alive and thus there were no significant differences between injection groups (Figure 3a). At 24 h, the percent survival was significantly different between PBS and Alizarin GG, Alizarin R, and Olsalazine. Alizarin GG, Alizarin R, and Olsalazine were not significantly different from each other. The average survival after 24 h of 900 nL PBS-injected mosquitoes was 100% with a standard deviation of zero. Alizarin GG was 74.0% with a standard deviation of 16.5, Alizarin R was 72.0% with a standard deviation of 16.9, and Olsalazine was 80.0% with a standard deviation of 13.3 (Figure 3b). Because all the mosquitoes were still alive at the 2 h time point, the stress of the injection wounding was insufficient to kill the mosquitoes. Furthermore, the additional stress of volume-challenging is insufficient to kill mosquitoes, as evident by the PBS mosquitoes’ survival at 24 h.

The addition of a dye is mildly toxic to mosquitoes in the 900 nL injections. The toxicity was slightly greater in both Alizarin dyes than Olsalazine, although not significantly. When viewed in combination with the 100 nL injections, the added toxicity is attributable to the total dye by mass, not the concentration. Because we saw significant differences in mortality in the 900 nL injections compared to the 100 nL injections, we concluded that 900 nL would be an appropriate volume to use for volume-challenging and gene expression experiments, as moderate mortality shows that the dyes are affecting normal mosquito processes while not killing them outright. Because all of the mosquitoes under these stresses survive beyond 2 h, we utilized those experimental conditions to investigate how they ameliorate the challenge of xenobiotic stress.

### 3.3. Urinalysis

When we volume-challenged mosquitoes with PBS, they responded by voiding roughly half the injected volume in 2 h. Mosquitoes injected with Olsalazine responded similarly by voiding almost half the injected volume, ~6% less of the volume than PBS-injected mosquitoes. Mosquitoes injected with Alizarin GG voided roughly one-third of the volume injected, a ~44% decrease compared to PBS. Mosquitoes injected with Alizarin R voided roughly one-sixth of the volume injected, a ~69% decrease compared to PBS (Figure 4a). Comparison of voided volume was performed using one-way ANOVA and Tukey’s HSD. While there was a large range for each round of experiments, the pattern within each round was consistent, wherein PBS and Olsalazine almost always had greater excretion volumes than Alizarin GG, which was then almost always greater than Alizarin R.

The concentration of dye excreted was greatest for Olsalazine, which is roughly three times more concentrated than Alizarin GG and six and a half times more concentrated than Alizarin R (Figure 4b). This aligns with the trends we see for voided volume. While the average excretion concentration for Olsalazine is 2.4 mM, some experimental groups had excretion concentrations as high as 3 mM which is higher than the injected concentration (2.5 mM), suggesting concentration of the solutes during excretion. Olsalazine was the only dye in which the excretion concentrations surpassed the injection concentration (Figure 4b).

Similarly, the percentage of dye removed by the mosquito is by far the greatest for Olsalazine, followed by Alizarin GG, then Alizarin R at 48%, 10%, and 2.8%, respectively. Alizarin GG and Alizarin R are both significantly less than Olsalazine, whereas Alizarin GG is not significantly less than Alizarin R (Figure 4c). The lack of a full percentage of the dye being voided by the mosquito suggests some of the dye could be chemically modified by the mosquito’s detoxification pathways, such as through Cytochrome P450 (CYP) enzymes [25]. To explore that possibility, we examined the absorbance spectra of the voided dyes.

We collected spectra from 175 to 800 nm initially to find the maximal absorbance and determine the excreted concentration using the molar absorptivity, but a reexamination allowed us to evaluate molecular changes to the dyes that might alter their spectra. We saw no evidence that the dye is being altered in any way before excretion within the ultraviolet and visible regions. That is, the absorbance peaks of the dye found in the urine are not different from the injected dye (Figure 5). Therefore, we believe that the dye we are injecting remains intact when excreted, and no detoxification products are present that absorb in the ultraviolet or visible regions of the urine.

### 3.4. Gene Response to Dye/Volume—Challenge

Following the changes in the rate of dye clearance, we next set out to determine if the putative OCT and OCTN transporter genes responded to alleviate the stress. We set the gene expression level from the 2 h time point with 900 nL PBS (volume-challenge) as the baseline and determined the relative difference for each dye. At 2 h, expressions of *AAEL012443* and *AAEL004479* decreased slightly in response to Alizarin dyes, while *AAEL024953* and *AAEL026837* increased slightly in Olsalazine-treated mosquitoes. *AAEL004479* expression decreased significantly for Alizarin GG (*p*-value = 0.0388). *AAEL004451* expression was, on average, increased, but the variability across biological replicates was very high, and *AAEL000902* did not see any notable changes in expression (Figure 6, top row).

When we compared gene expressions at 24 h to our baseline, several differences were present, but again, most changes remained subtle. For Alizarin R and Olsalazine, there was a shared decrease in *AAEL004451* expression. *AAEL012443* expression increased slightly after 24 h in Alizarin R-injected mosquitoes, similar to that of PBS-injected. *AAEL000902* expression was decreased only by Olsalazine. *AAEL024953* expression increased slightly for Alizarin GG and increased more for Alizarin R, but not significantly from the baseline. *AAEL026837* increased in Olsalazine, with one data point having an eight-fold increase in expression compared to PBS 2 h, and while the other data points were increased, it was not beyond what could be explained by chance. Interestingly, *AAEL004479* decreased across all injection groups, including PBS 24 h, and was significantly different between PBS- and Alizarin GG-injected mosquitoes at 24 h (*p*-value = 0.0202) (n = 3) (Figure 6, bottom row).

In addition to the comparison between treatment conditions recounted above, we also noted common trends when comparing the abundance of our genes of interest relative to the housekeeping genes. *AAEL000902* and *AAEL024953* show the highest levels of expression throughout each treatment (time or injection composition). *AAEL012443* and *AAEL026837* have the next highest levels of expression, followed by *AAEL004479* and *AAEL004451*.

## 4. Discussion

Insects have evolved a robust system for dealing with xenobiotics [26], which involves not only the transport of foreign molecules, but also the detoxification of molecules that are resistant to removal. In the female mosquito’s life, there are two clear times in which they are subject to secondary metabolites, first during nectar feeding when plant attractant or defensive molecules are ingested, and second when they take a blood meal [27]. The transport can be thought of as having two parts: the first is the movement of molecules into the epithelial cells, and the second is the removal of those molecules from cells. A great deal of work has focused on the latter partition of this process by examining the ABC transporters, whereas very little is known about the import of xenobiotics into epithelial cells [28,29,30], despite their presence being demonstrated in several insects [31,32,33,34].

Here, we have utilized a novel set of xenobiotics to explore their removal by the mosquito. However, this is not the first instance of dyes being utilized as xenobiotics in the investigation of transport proteins in insects [33,34,35,36,37] nor of measuring the volume of urine produced when mosquitoes are volume-challenged [38,39,40,41]. Nevertheless, to the best of our knowledge, this is the first example where a series of structurally similar dyes was evaluated for their effect on urine volumes.

If all things were equal, one could assume that the physiological mechanisms responsible for dealing with the volume-challenge and the xenobiotic challenge would function independently of each other, the first working to maintain water and salt balance while the second removing the xenobiotics quickly and effectively. However, our results suggest that these two processes are closely interconnected, where the volume of urine depends upon the xenobiotic molecule and not the initial concentration.

In Figure 4b, one interesting finding is the number of times the concentration of Olsalazine in the urine exceeded that which was injected (represented as the dotted line at 2.5 mM). The urine may be more concentrated because of the reabsorption of water in the hindgut prior to urine excretion [42]. Evidence of reabsorption elucidates two opportunities for activating transporter expression: the actual movement of Olsalazine throughout the mosquito and the reabsorption of metabolites with water prior to dye excretion. This suggests that there is something unique about Olsalazine, not shared by Alizarin dyes.

One possible explanation is the ionizability of each dye; that is, Alizarin GG and Alizarin R consist of two ions, the negatively charged dye and a sodium counterion. In contrast, Olsalazine consists of three total ions as the dye has two carboxylates balanced by two sodium cations (Figure 2). The addition of one ion could contribute to Olsalazine being excreted more readily; this is because mosquitoes produce urine in the Malpighian tubules by actively transporting ions into the tubule lumen, which creates a concentration gradient where water passively follows through membrane channels called aquaporins [43,44,45]. Because Olsalazine has more ions and a greater molality, the mosquito can exert a greater force to move water into the Malpighian tubules, leading to greater excretion volumes (as seen in Figure 4a). However, the molality difference fails to explain the variation in excretion volumes of Alizarin GG- and Alizarin R-injected mosquitoes, both of which have the same number of ions.

The additional hydroxyl group of Olsalazine is another polarized bond that is capable of hydrogen bonding and thereby greatly increases the solubility of Olsalazine in water compared to the Alizarins. This is evident when looking at the log10 of the partition coefficient (LogP) values of each dye. The partition coefficient is the ratio of a compound’s concentration in an organic phase relative to its concentration in an aqueous phase at equilibrium. The lower the LogP, the more soluble the dye is in water or saline. The LogP values for our dyes are 0.51 for Alizarin R, 0.48 for Alizarin GG, and −0.17 for Olsalazine (Molinspiration Cheminformatics. Available online: https://www.molinspiration.com, accessed on 1 April 2025). These match our observations that Olsalazine is much more soluble in PBS than both Alizarin dyes, and that Alizarin GG is measurably more soluble than Alizarin R. With the addition of LogP, the concentration of Alizarin R in the urine is reasonable because Alizarin R is more likely to be retained within the hemolymph either as a precipitate or by sequestration in lipid-rich tissues, such as body fat. The stain Nile Red (LogP = 4.58) is utilized to quantify the amount of lipids in the body fat of mosquitoes [46]. The limited amount of Alizarin R in the hemolymph therefore reduces the number of ions being translocated to the tubule lumen. This reduces the concentration gradient and the passive water that flows, and ultimately the amount of urine produced.

Although Olsalazine was excreted most rapidly, none of the groups fully excreted the dye at 2 h post-injection. In fact, Alizarin GG and Alizarin R only removed ~5% and ~3%, respectively (Figure 4c). In the period between 2 and 24 h, the mosquitoes then successfully excreted more of the remaining dye, as evidenced by the presence of dye-colored spots in the recovery habitats over the 24 h following the volume challenge. We observed the greatest number of spots for Alizarin R, followed by Alizarin GG, and lastly by Olsalazine, which suggests that Alizarin R is being processed by the mosquito, but at a slower rate. When visually examining the amount of urine droplets on the collection tubes after 2 h, it is inversely related to what is observed in the environmental chambers after 24 h. For Olsalazine, droplets would typically appear after around 15 min, while for Alizarin GG and Alizarin R, droplets wouldn’t appear until around 60 min and 90 min, respectively. This order of dye appearance and urine volume supports the model where active organic ion transport is connected to the passive movement of water. Our data suggests that a mechanism exists where the mosquitoes reserve or recycle PBS while working to translocate higher LogP xenobiotics like the Alizarin dyes. This supports the interconnectedness between xenobiotic transport and urine production.

The mosquitoes clearly process and excrete the dyes differently, leading us to hypothesize that the putative transporters AAEL004451, AAEL012443, AAEL000902, AAEL024953, AAEL026837, and AAEL004479 may mediate these differences, but the gene expression data under our conditions did not indicate an outsized role for any of them. However, several subtle changes were observed and may hint at minor responses to xenobiotics and volume-challenging.

An interesting result was the consistent decrease in *AAEL004479* in volume-challenged mosquitoes after 24 h, regardless of the dye injected (Figure 6). This may indicate the existence of a common mechanism of response to metabolic or volumetric success that downregulates *AAEL004479*. Our findings are consistent with past studies where *AAEL004479* does not increase in a whole-body mosquito after a blood meal [47]. Additionally, *AAEL004479* is most highly expressed in the crop [48]. Because we injected our dyes into the hemolymph, the crop would not be involved in the processing and excretion of the dye. In other words, because *AAEL004479* is predominantly expressed in an organ dedicated to sugar feeding, it may be downregulated during volume challenge, which simulates blood feeding. This downregulation could save resources and energy for other processes that are essential to the mosquito’s survival under volume stress.

While *AAEL004479* expression was lower in every condition, *AAEL004451* appears to only decrease in expression at 24 h in Olsalazine and Alizarin R-injected mosquitoes and is highly variable in all other conditions compared to PBS at 2 h (Figure 6). This may suggest that *AAEL004451* does not play an outsized role in the clearance of xenobiotics in the mosquito. However, the large variation in our results could also be attributed to the location of *AAEL004451* expression in the mosquito. Past studies have found that *AAEL004451* expression is low in whole-body mosquitoes fed with sucrose but extremely high in the hindgut in these same mosquitoes [48]. Furthermore, *AAEL004451* increases in expression in whole-body mosquitoes after a blood meal, which further supports that it is an important transporter to study [47]. Therefore, our finding for AAEL004451 should be viewed narrowly within these experimental conditions; it is likely that isolating the hindgut in future experiments could provide more consistent results and a clearer picture of changes in *AAEL004451* expression. *AAEL012443* has also been found to have high expression in the hindgut [48] and could benefit from similar experiments.

*AAEL000902* expression was relatively unchanged over the experiments. Because *AAEL000902* is known to be highly expressed in the Malpighian tubules [47], it would seem a likely candidate to filter xenobiotics from the mosquito’s bodily fluid. The notion that it would be responsive is supported by literature that found AAEL000902 increases 4–8 h after a blood meal in the midgut [49]. Our data shows only a slight increase, suggesting that it is not responsive to Alizarins or Olsalazine.

A broader takeaway from our expression data can be made when comparing the transporter expression levels to the housekeeping genes *ACT* and *RPS17*. Here, *AAEL000902* and *AAEL024953* exhibited the highest baseline expression, which varied little across treatment times and conditions, suggesting their roles support essential functions and are less responsive to xenobiotic stress. To evaluate whether these genes may respond to facilitate volume management, we performed a preliminary assay in anesthetized, needle-wounded mosquitoes and analyzed the transcript levels at 2 h. This single replicate revealed no immediate or clear expression differences from volume-challenged mosquitoes (Appendix A). This again supports the notion that they are essential, but for a function unresponsive to xenobiotic or volume stress.

While we do see changes in gene expression, none of these changes are drastic, and they generally lack consistency. While it is possible that these genes are simply not responding strongly to the exposure to xenobiotics, the more likely explanation is that subtle changes in expression may have been overshadowed by the large pool of RNA from the whole body of the mosquito. Therefore, in order to fully elucidate these transporters, we need three adaptations to our approach. First, repeating these experiments with mosquitoes dissected into the major components of their digestive and excretory system, including the midgut, hindgut, Malpighian tubules, crop, and ovaries, can provide context for where exactly the observed changes in gene expression are occurring. Determining the tissue-specific expression can yield insights into the function of each putative OCT(N) and allow us to connect transcriptional patterns with physiological purpose in a more mechanistic and interpretable way. For instance, if we see increased expression only in the Malpighian tubule, it would suggest a role in the translocation of xenobiotics to the urine, whereas expression in the hindgut would point to a role in reabsorption of metabolites. Similarly, if we find putative transporters respond antithetically in digestive versus non-digestive tissues, this may indicate a systemic response to xenobiotic stress, but if the division is not cleanly split, a more complex response is likely. Additionally, using RNA interference to limit the transporter expression and repeating the xenobiotic and volume challenges may reveal how critical the transporters are to the response. Lastly and most definitively, functional assays of the transporters in oocytes can characterize the activity and selectivity of the predicted OCT(N)s. Until such validation is performed, the role of these transporters in xenobiotic clearance remains putative.

## 5. Conclusions

To date, the mechanisms and key players underlying *Ae. aegypti* xenobiotic transport by OCT(N) transporters remain largely uncharacterized, but this study begins the search to identify which OCT(N)s are most greatly contributing to xenobiotic clearance and provides a basis for future experiments. More significantly, this work provides evidence of links between xenobiotic transport and urine production, a point that may be useful in developing new insecticide targets. Because of the large populations affected by vector-borne illness, even modest advances could meaningfully reduce global disease burden. Additionally, developing insecticides that are environmentally friendly, vector-specific, and economic can help make mosquito-borne illness prevention more accessible for the countries and people who need it the most.

## Figures and Tables

**Figure 1 insects-16-01196-f001:**
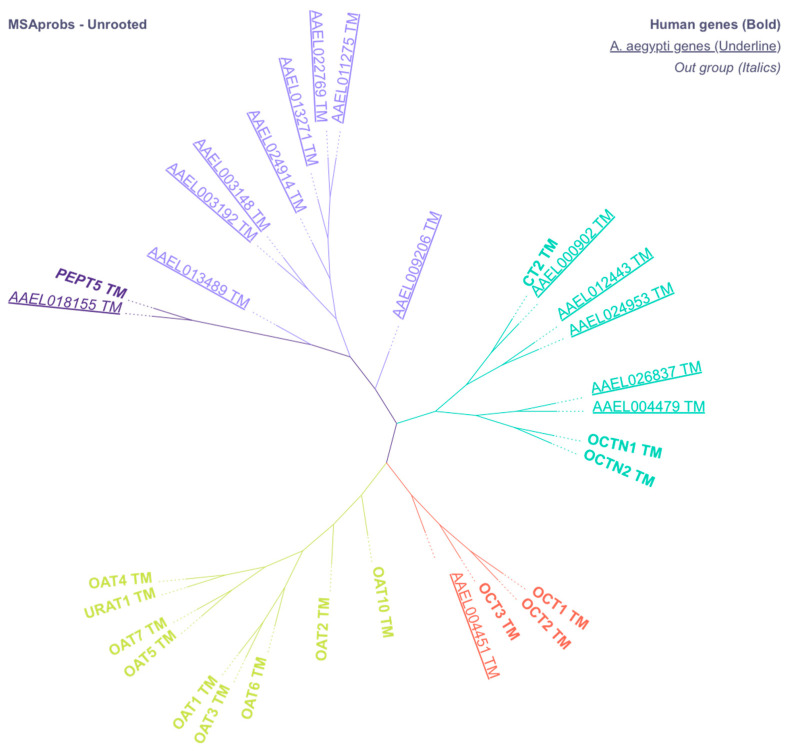
Unrooted phenogram of putative genes coding for mosquito OATs, OATPs, OCTs, and OCTNs (underlined text) aligned with human transporters (Bold text). All putative *Ae. aegypti* sequences and human sequences were analyzed for transmembrane regions and truncated to remove N- and C-termini. PEPT5 was used as a control in our analysis, and branch length was ignored.

**Figure 2 insects-16-01196-f002:**
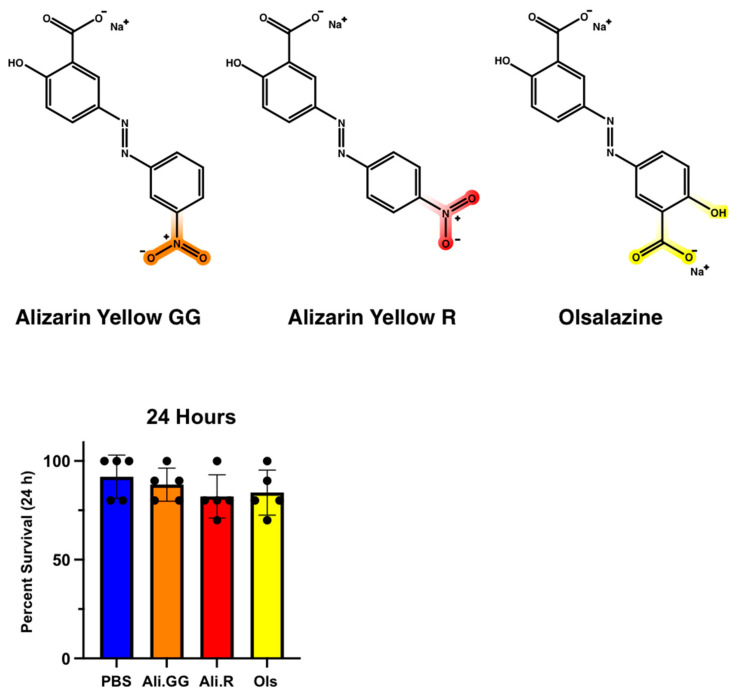
(**Top**): Structure of (**left**) Alizarin Yellow GG (NaC_15_H_13_N_3_O_5_), (**center**) Alizarin Yellow R (NaC_13_H_9_N_3_O_5_), and (**right**) Olsalazine (Na_2_C_14_H_8_N_2_O_6_) (PubChem). (**Bottom**): Percent survival of mosquitoes injected with PBS (control), Alizarin Yellow GG, Alizarin Yellow R, and Olsalazine at 24 h with a 100 nL injection. Dots represent a single trial of 10 mosquitoes. (n = 5 trials).

**Figure 3 insects-16-01196-f003:**
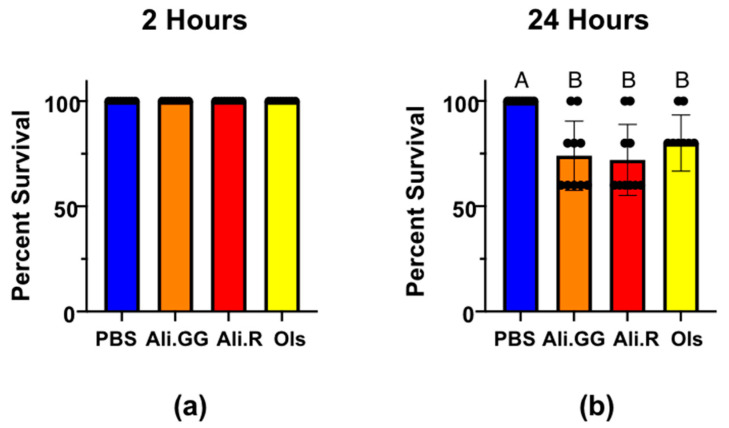
Percent survival of mosquitoes injected with PBS (control), Alizarin GG, Alizarin R, and Olsalazine at (**a**): 2 h with a 900 nL injection, and (**b**): 24 h with a 900 nL injection. Different letters over the bars indicate a significant difference at *p* < 0.05 (Tukey’s HSD). Dots represent a single trial of 10 mosquitoes (n = 10).

**Figure 4 insects-16-01196-f004:**
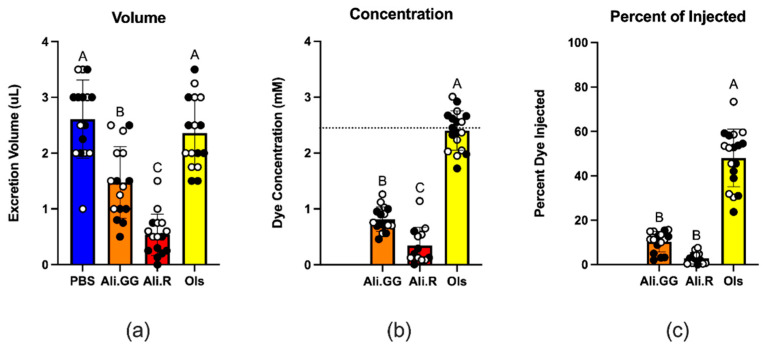
(**a**): Cumulative excretion volume of five mosquitoes injected with 900 nL of PBS (control), Alizarin GG, Alizarin R, and Olsalazine after 2 h. (**b**): Excretion concentration (mM), where the dashed line represents injected dye concentration (2.5 mM). (**c**): The percent of injected dye that was excreted per mosquito after 2 h. Circles represent a single trial of 10 mosquitoes, closed circles indicate mosquitoes utilized for mortality, and open circles indicate mosquitoes harvested for RNA. Different letters over the bars indicate a significant difference at *p* < 0.05 (Tukey’s HSD) (n = 16).

**Figure 5 insects-16-01196-f005:**
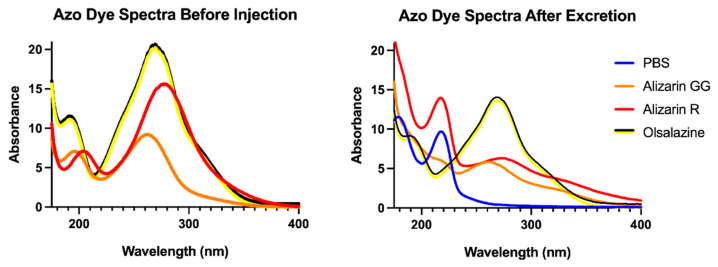
Absorbance spectra of Alizarin Yellow R, Alizarin Yellow GG, and Olsalazine before injection (**left**) and after excretion with PBS (**right**).

**Figure 6 insects-16-01196-f006:**
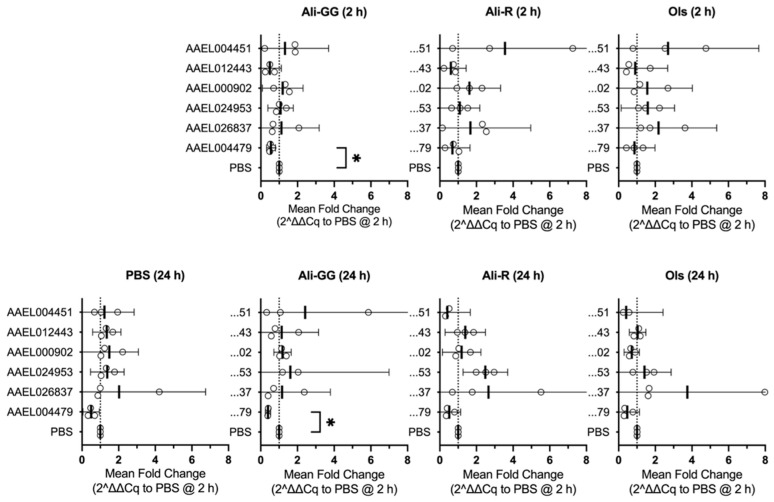
Mean fold change in gene expression at 2 h (**top row**) and at 24 h (**bottom row**) relative to 2 h PBS expression. Open circles are the mean of the technical replicates, heavy lines represent the mean of the biological replicates, and the error bars are the 95% CI. Star denotes *p*-value < 0.05. (n = 3).

## Data Availability

The data that support the findings of this study are available from the corresponding authors upon reasonable request.

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
