# Peer review of "Physiological and Putative Organic Cation Transporter Expression Response to Alizarin Dye Exposure in Aedes aegypti Mosquitoes"

_insects, 2025, doi:10.3390/insects16121196_

Round 1

Reviewer 1 Report

Comments and Suggestions for Authors

The manuscript by Kennel and Rouhier investigates a highly relevant and underexplored area in vector biology: the role of putative organic cation transporters (OCTs/OCTNs) in xenobiotic clearance in Aedes aegypti. The study is timely, as understanding these molecular mechanisms could lead to novel vector control strategies. The experimental approach, combining physiological assays (urinalysis, mortality) with gene expression analysis, is appropriate and well-described. The authors have identified a clear, dye-specific physiological response in terms of urine volume and clearance rate.

However, the manuscript in its current form has significant weaknesses that prevent full endorsement for publication. The primary issue is the overinterpretation of the qPCR data, which shows only subtle and largely inconsistent changes in gene expression. The conclusions drawn from these minimal expression changes are not sufficiently supported by the data presented. Furthermore, the lack of functional validation for the identified transporters and the use of whole-body RNA samples dilute the potential impact of the findings. Addressing the points below, particularly by tempering conclusions and providing a more critical discussion of the limitations, would significantly strengthen the manuscript.

Major Comments

Overstated Gene Expression Results and Conclusions: The central claim of the study is that xenobiotic exposure alters transporter expression profiles. However, the data in Figure 6 largely do not support strong conclusions. Most changes are subtle (often less than 2-fold), lack statistical significance, and show high variability. The manuscript would be greatly improved by a more cautious and critical interpretation of these results. For instance, stating that the dyes "alter the transporter expression profiles" is an overstatement; a more accurate description would be that they "induce minor and variable changes in expression." The discussion often speculates on the role of specific genes based on these minimal changes, which should be framed as hypotheses for future work rather than firm findings.

Lack of Functional Validation: The study identifies putative transporters based on sequence homology but provides no direct evidence of their function. Do these proteins actually transport the tested dyes or other organic cations? The gene expression data alone are correlative. A major improvement would be to include a functional assay, even if preliminary (e.g., heterologous expression in Xenopus oocytes or a cell culture model to test dye transport). Without this, the link between expression changes and functional xenobiotic transport remains speculative.

Use of Whole-Body RNA: As the authors correctly note in the Discussion (Lines 491-496), the use of whole-body homogenates likely masks tissue-specific expression changes that are critical for understanding excretory and detoxification processes. The most compelling findings from other studies (e.g., high expression of AAEL004451 and AAEL012443 in the hindgut) are cited, yet this methodological choice prevents the current study from building clearly upon that foundation. The recommendation to repeat experiments with dissected tissues (Malpighian tubules, hindgut, midgut) is excellent and should be highlighted as a essential next step, but its mention also underscores a key limitation of the present data.

Statistical Power and Replicates: The gene expression experiments were performed with n=3 biological replicates. Given the high variability observed in some datasets (e.g., AAEL026837), this low n-value may be insufficient to draw reliable conclusions. The authors should explicitly state this as a limitation and consider performing additional replicates to bolster their claims, or further temper the conclusions based on the existing data.

Minor Comments

Introduction: The introduction is well-written and sets a clear context. However, the final paragraph (Lines 74-80) could be slightly refined. The connection between a blood meal (0-72h) and the 2h/24h time points chosen for a hemocoel injection of dye is somewhat tenuous. Justifying the 2h and 24h time points more directly in the context of the initial diuretic response and subsequent recovery/processing phase would be helpful.

Figure 4: Panels 4a and 4c are clear. In Panel 4b, it is very striking that the excreted concentration of Olsalazine often exceeds the injected concentration. The discussion of this point (Lines 367-371) is good, but it could be expanded. Could this be due to significant water reabsorption in the hindgut concentrating the urine? This seems to be a key piece of physiological evidence that deserves more emphasis.

Discussion on LogP: The discussion linking LogP values to excretion patterns (Lines 387-401) is insightful and one of the manuscript's strengths. This provides a plausible and testable mechanistic hypothesis for the observed differences in dye handling.

Data Presentation:

Figure 1: The phylogenetic tree is central to the study. The figure legend should explicitly state the method used for tree construction (such as Maximum Likelihood, Neighbor-Joining) and what the branch lengths/nodes represent.

Figure 6: The graph is difficult to read. Consider using a different plot type (a dot plot with means) or separating the 2h and 24h data into two panels for clarity. The use of striped bars for the 24h time point is not ideal.

The manuscript is generally well-written. A few minor points:

Line 31: "...alter the transporter expression profiles, the volume and composition..." -> Consider rephrasing for clarity, e.g., "...alter the transporter expression profiles as well as the volume and composition..."

Line 360: "inexplicably linked" likely means "inextricably linked."

Line 416: "reserve or recycle PBS" to"reserve or recycle water/saline."

Reviewer 2 Report

Comments and Suggestions for Authors

General Comments

The manuscript submitted by Kennel and Rouhier on organic cation transporter expression responses to dyes is interesting because it explains the expression of genes involved in xenobiotic metabolism. This paper will be useful for the scientific community since alternative tools to chemical insecticides are actively being researched. However, the quality of the manuscript should be improved before publication. For instance, some sentences are repeated, others need clarification, a few orthologs and their functions should be included in the discussion to support predictions, and there is excessive use of personal pronouns throughout the manuscript.

Please find below detailed suggestions to improve the quality:

Specific Comments

  • L17: Replace mosquito carried disease with mosquito-borne diseases or mosquito-borne pathogens. Mosquitoes do not carry diseases; they transmit pathogens, and diseases occur in vertebrates.
  • Abstract: Reduce the use of personal pronouns such as “we.” For example, L28 could be revised to: “mRNA expression was analyzed…”
  • L42: Replace harmful diseases with pathogenic viruses and modify the sentence accordingly, since West Nile virus is a pathogen, not a disease. Technically, mosquitoes are vectors of pathogens, not diseases.
  • L65–67: The sentence appears incomplete; include the consequence of silencing chitin synthase.
  • L73–77: This is a very long sentence; split it into two for better readability.
  • L86: Replace by with with since you used the software to align your sequences.
  • L95: Remove The before Aedes aegypti. Species names do not require articles.
  • L109: Confirm if you used 1X PBS; if correct, please specify.
  • L115: Include the company name where you purchased your filter in parentheses.
  • L116–117: Clarify whether you performed three biological replicates or three technical replicates.
  • L129 & L152: Add the temperature for centrifugation steps.
  • L154–156: The RNA isolation section lacks details such as concentrations and volumes. You can simplify by stating that you followed the manufacturer’s guidelines or by citing a reference.
  • L163–164: Include accession numbers for ACT and RPS17 or cite one or two relevant papers.
  • L183–184: Include the concentration of your qPCR master mix.
  • L192–195: Add a citation for the ΔCt method.
  • L198–200: This sentence duplicates L136–137; delete it.
  • Figure 2: Include “24 h” at the top of the graph as in Figure 3 to facilitate comparison between 100 nL and 900 nL.
  • L403–404: The sentence appears incomplete; verify and revise.
  • Discussion: Reduce the use of personal pronouns such as “we.” Also, include the functions of orthologous genes to support your predictions.
  • L486: Define TCA (Tricarboxylic Acid) for clarity, as non-biochemists may not recognize the abbreviation.
Comments on the Quality of English Language

The authors should verify the appropriate use of prepositions (by, on and others) and articles. In addition, they should reduce the use of personal pronouns, especially 'we'.

Reviewer 3 Report

Comments and Suggestions for Authors

This work presents comprehensive research on the mechanisms of xenobiotic transport in Aedes aegypti, with particular emphasis on the identification and functional analysis of potential organic cation transporters (OCT/OCTN). This topic is highly relevant and has significant practical implications, as understanding the mechanisms of metabolite and toxic compound transport in mosquitoes may enable the development of new strategies for controlling infectious disease vectors. The authors employed an interdisciplinary approach, combining bioinformatics analyses, physiological tests related to diuresis and excretion, and gene expression assessment using qPCR. This combination of methods allows for a coherent assessment of the potential role of transporters in mosquito metabolism and detoxification.

The strengths of this work include, above all, the comprehensive bioinformatics analysis, which utilized a variety of tools for sequence comparison and transmembrane domain prediction, allowing for the reliable identification of candidate genes. The maintenance of A. aegypti colonies under well-controlled environmental conditions, which limits the influence of external factors on the obtained results. The authors presented a detailed description of the experimental procedures, including xenobiotic injection, toxicity measurements, and analysis of excretion volume and composition. The use of two injection volumes (100 nL and 900 nL) was an effective solution, enabling the distinction between toxic effects and volumetric stress. The high quality of the molecular analyses is confirmed by carefully designed primers, their efficiency tests, and the inclusion of appropriate controls (No RT, NTC). Another significant strength of the work is its consistent interpretation – the authors linked physiological results with gene expression, allowing for a more comprehensive understanding of the mosquito's response to xenobiotic exposure. However, there are some shortcomings. The criteria for selecting gene sequences were not sufficiently explained – the established e-value threshold (e-30) was not justified, making it difficult to assess the reliability of gene selection. The chapter on methodology is overly extensive in places and could be supplemented with a flowchart illustrating the experimental procedure, which would improve its clarity. It is also important to clarify whether the reported n=3 values ​​refer to biological or technical replicates, as this is important for statistical interpretation. Although the authors used ANOVA and Tukey HSD tests, no mention was made of checking for normality of distribution or potential corrections for multiple comparisons, which may affect the reliability of the results obtained with small sample sizes. The study results are consistent and provide a solid starting point for further analyses. The identification of 14 potential genes encoding organic cation transporters, six of which were selected for detailed study, is a valuable achievement. The results of toxicity tests and excretion measurements correspond well with gene expression analysis. However, the observed changes in mRNA levels are moderate, which may suggest an adaptive rather than a severe stress response. An interesting observation is the decrease in the expression of some genes, including AAEL004479, regardless of the type of xenobiotic used, which may indicate the existence of common mechanisms of response to metabolic or volumetric stress. From a practical perspective, these studies could provide a basis for developing new methods for limiting the spread of insect-borne diseases. To strengthen these conclusions, it would be advisable to supplement the experiments with functional confirmation of the activity of the studied genes, for example, using RNAi methods or proteomic analyses.

Round 2

Reviewer 1 Report

Comments and Suggestions for Authors

I would like to thank the authors for their thorough and thoughtful responses to my initial comments. They have made substantial revisions to the manuscript, addressing several key concerns raised during the first round of review.

While the authors have adequately addressed most of the concerns, a few points may still benefit from minor clarification or emphasis:

The authors acknowledge that functional validation (heterologous expression or RNAi) is beyond the scope of this study. However, the discussion could more explicitly highlight that the role of these transporters in xenobiotic clearance remains putative until such validation is performed.

Although the use of whole-body RNA is now clearly framed as a limitation, the authors may wish to briefly note in the Discussion how future tissue-specific studies could resolve key mechanistic questions.

Correction and Clarification:

In my previous review, I suggested a text change on what was then Line 416 ("reserve or recycle PBS" to "reserve or recycle water/saline"). Upon reflection, the authors' original use of "PBS" was precise in the context of their experimental method. Please disregard that specific suggestion and retain the original phrasing.
